# Optimization of the Process for Green Jujube Vinegar and Organic Acid and Volatile Compound Analysis during Brewing

**DOI:** 10.3390/foods12173168

**Published:** 2023-08-23

**Authors:** Guifeng Li, Ni Yan, Guoqin Li, Jing Wang

**Affiliations:** 1College of Food Science, Shanxi Normal University, Taiyuan 030031, China; 2Modern College of Humanities and Sciences, Shanxi Normal University, Linfen 041000, China

**Keywords:** green jujube vinegar, response surface optimization, organic acid, volatile compounds, HS-SPME-GC-MS

## Abstract

Healthy fruit vinegar has become very popular recently in China. This study aimed to produce fruit vinegar with a good taste, high nutritional value, and strong functional properties from green jujube. This study investigated the optimization of the process for green jujube vinegar using response surface methodology. The optimum fermentation parameters for green jujube vinegar were determined as follows: initial alcoholicity 6%, acetobacter 8%, fermentation temperature 32 °C, and time 7 d. The organic acids of the optimized sample were evaluated by HPLC, and the volatile substances were identified and analyzed by HS-SPME and GC-MS during the fermentation and aging of the green jujube vinegar. The results showed that the variation trends of the different organic acids during the making of the green jujube vinegar were significantly different. Organic acids are the key flavor compounds of green jujube vinegar, and their changes were mainly attributed to microbial metabolism. In particular, the green jujube vinegar stood out in terms of volatile aroma compounds, including a total of 61 volatile compounds whose major components were acetic acid, isoamyl acetate, ethyl acetate, 3-hydroxy-2-butanone, methyl palmitate, and ethanol. The results can provide theoretical support for the production of green jujube vinegar.

## 1. Introduction

Vinegar is one of the most widespread condiments globally and plays an important role in the diet, encouraging calcium absorption and stimulating appetite. At the same time, vinegar, which is known as the “fourth-generation beverage” [1,2], is very beneficial to human health due to its antioxidant, anticarcinogenic, and antibacterial properties, its ability to lower cholesterol and blood pressure, and its role in preventing cardiovascular disease [3,4,5]. Acetic acid is an essential and major constituent of vinegar; it has beneficial effects against hypertension, hyperglycemia, and dyslipidemia [6,7]. Therefore, vinegar has long been used as a traditional medicine to regulate blood pressure and blood glucose. In addition to acetic acid, vinegar contains various functional compounds derived from its raw material, such as phenolics, flavonoids, superoxide dismutase (SOD), and vitamin C [8,9].

Jujube (*Ziziphus jujuba* Mill.), a fruit of the Rhamnaceae family, is extensively cultivated and has a 4000-year-long history in China [10]. The jujube fruit is one of the most widely consumed fruits in Asia. It is rich in a variety of biologically active ingredients and has high nutritional value and medicinal value according to both traditional medicine and modern scientific research [11,12]. The nutritional compositions and health functions of the jujube fruit vary according to cultivation method, variety, and maturity [13]. The maturity of jujubes was divided into three grades: white maturity, half-red maturity, and red maturity [14]. The fruit at the white maturity level is the green jujube. Some authors focused on the active components and biological functions in the three maturity levels of jujube [15]. The results showed that green jujube has the highest total phenolic content (TPC), total flavonoid content (TFC), total phenolic acid contents, organic acid levels, and vitamin C levels [14]. These bioactive compounds feature multi-functional properties, such as lowering blood pressure, blood lipids, and antioxidants and anticarcinogenic and antimicrobial activities [16,17]. It is evident that the white-maturity green jujube has the highest utilization value.

At present, the planting area of jujube trees in China has increased, and the output has increased year by year. However, its low degree of industrialization and lack of market competitiveness have led to a large number of unsalable dates, heavy losses for jujube farmers, and a severe dilemma for the jujube industry. Therefore, it is necessary to adopt new ideas and develop new products to overcome the downturn of the jujube industry. However, so far, no studies have been conducted on the processing and features of the vinegar produced from green jujubes. Therefore, it is important to process green jujubes into vinegar, which increases the bioactive compounds and biological activities of the fruit. This study aimed to produce vinegar with strong functional properties from green jujubes and to convert this fruit into a high-value-added product. The total acidity content has a direct impact on the quality of fruit vinegar and is an important indicator with which to verify the good character of fruit vinegar in China. The volatile compound was the direct influencing factor in the sensory evaluation of the fruit vinegar and was also an important index in the evaluation of the quality of the fruit vinegar. The changes in volatile components in fruit vinegar are usually related to a series of chemical reactions and microbial metabolism during fermentation and aging. For this purpose, this research systematically optimized the fermentation process of green jujube vinegar and investigated the changes in various organic acids and volatile compounds in optimized green jujube wine during winemaking, which provides the scientific basis for the making of green jujube vinegar.

## 2. Materials and Methods

### 2.1. Preparation of Green Jujube Juice

Green jujubes (*Chinese winter jujube*) were collected from Yao Xiang jujube planting base, Linfen City, Shanxi Province, China. The average content of pectin in jujube was 204.74–682.16 mg/100 g. After cleaning and removing the jujube pits, the green jujubes were softened in a water bath at 90 °C for 15 min at a fruit-to-water ratio of 1:2, after which 0.3% pectinase was added and hydrolyzed at 40 °C for 3 h. Green jujube juice was obtained by the juicer pressing method and strained through gauze (soluble solid content, SSC, 11 °Brix, pH 3.81) [18].

### 2.2. Fermentation

Alcoholic fermentation was carried out in 5.0 L glass containers containing 3.0 L green jujube juice. Dry yeast (*Saccharomyces cerevisiae*, Angel Yeast Co., Ltd., Yichang City, Hubei Province, China) was activated with 2% sugar water for 15–30 min at 35−38 °C, with a 1:5 ratio of dry yeast to sugar water. After adjusting the initial sugar content to 24%, adding 0.3% yeast and 80 mg/L SO_2_, the green jujube wine was fermented at 23 °C for 7 days. The evolution of fermentation was monitored by daily measurements of total sugar content with a handheld refractometer in the fermenter. The alcohol content was measured according to the method described by the national standards of China GB 5009.225-2016 (determination of ethanol concentration in wine, in Chinese).

After alcoholic fermentation, acetic acid fermentation was carried out. The alcohol content of the green jujube fermentation broth was adjusted to 6% with distilled water, after which 8% activated acetic acid bacteria (*Acetobacter pasteurianus*, Shanghai Difa Brewing Biology Co., Ltd., Shanghai, China) were added. During fermentation at 32 °C for 7 days, the total acidity was measured daily via the acid–base titration method. After fermentation, the green jujube vinegar was obtained by taking supernatant fluid and aging it for 2 months [19]. The total acidity was measured according to the method described by national standards of China GB/T 12456-2008 (determination of total acid in foods, in Chinese) [20]. The determination of total acidity is based on the principle of acid–base neutralization. The acid is titrated with alkali solution, and the titration endpoint is determined with Phenolphthalein as indicator. The total acid content is calculated according to the consumption of lye.

### 2.3. Experimental Design

As a key parameter of green jujube vinegar, total acidity content plays an essential role in the acetic acid fermentation process. A preliminary investigation of the factors affecting the total acid content of fermented greengage vinegar was conducted using single-factor experiments, including inoculation amount, fermentation time, fermentation temperature, and alcohol content. The factors chosen were inoculation amount (4, 6, 8, 10, and 12%), fermentation time (4, 5, 6, 7, and 8 d), fermentation temperature (28, 30, 32, 34, and 36 °C), and alcohol content (4, 5, 6, 7, and 8%).

To obtain high-quality green jujube vinegar, a central composite experimental design with twenty-nine treatments was used to optimize the fermentation conditions (acetobacter inoculation amount, fermentation temperate, fermentation time, and initial alcohol content). Twenty-nine treatments were performed according to the Box–Behnken experimental design principles with 4 factors and 3 levels for each variable using Design-Expert 8.0. The response surface methodology (RSM) was employed to optimize the fermentation conditions [21]. The independent variables applied in the experimental design were inoculation amount (6, 8, and 10%), fermentation time (6, 7, and 8 d), fermentation temperature (30, 32, and 34 °C), and alcohol content (5, 6, and 7%), consistent with the coded levels (−1, 0, and 1), (−1, 0, and 1), and (−1, 0, and 1), as shown in Table 1.

### 2.4. Analysis of Organic Acids

#### 2.4.1. Sample Preparation

The organic acids of the optimized green jujube vinegar were studied during the brewing. Five vinegar-brewing stages were selected for organic acid analysis, including fresh jujube juice (0 d), alcohol fermentation (7 d), acetic acid fermentation (14 d), green jujube vinegar aged for 1 month (44 d), and green jujube vinegar aged for 2 months (74 d). In total, 10 mL of sample solution was taken and diluted with 40 mL methanol–water (1:1, *v*/*v*). After ultrasonic extraction (15 min) and centrifugation (5000 r/min, 10 min), the supernatant fluid was filtered through a hydrophilic filter membrane (0.22 µm) for organic acid analysis.

#### 2.4.2. HPLC Apparatus and Conditions

Seven organic acids were analyzed in a high-performance liquid chromatography (HPLC) system equipped with a chromatographic column (Acquity UPLC-HSS-T3, 2.1 mm × 100 mm, 1.8 µm, Waters, Milford, MA, USA) used for identification and analysis of organic acids at 210 to 230 nm in the ultraviolet detector [22,23]. Organic acid standards (1) oxalic acid, (2) tartaric acid, (3) lactic acid, (4) acetic acid, (5) citric acid, (6) malic acid, and (7) succinic acid were used for the identification and quantification of organic acids in green jujube vinegar.

After the pretreatment, samples were filtered through a 0.45 μm microporous membrane. The mobile phase consisted of 20 mmol/L potassium dihydrogen phosphate buffer (pH = 2) as A and acetonitrile as B. The program of gradient elution was injection volume 1.0 µL, flow rate 0.21 mL/min, column temperature of 35 °C, and 12 min gradient of mobile phase (0–2.5 min, 98%, A + 2% B; 2.5–4.0 min, 80% A + 20% B; 4.0–7.0 min, 60% A + 40% B; 7.0–8.0 min, 80% A + 20% B; and 8.0–12.0 min, 98% A + 2% B).

### 2.5. Analysis of Volatile Compounds

#### 2.5.1. Extraction of Volatile Compounds by Headspace Solid Phase Microextraction (HS-SPME)

The volatile compounds of green jujube vinegar were extracted via HS-SPME (75 µm, CAR/PDMS, Supelco, Bellefonte, PA, USA). Each green jujube vinegar sample (10 mL) was placed in a 20 mL SPME glass vial together with internal standard (10 µg, isoamyl phenylacetate) and kept for 30 min at 60 °C [24].

#### 2.5.2. Analysis of Volatile Compounds by Gas Chromatography-Mass Spectrometer (GC-MS)

The volatile compounds of green jujube vinegar were determined by GC-MS (6890N-5973, Agilent Technologies, Santa Clara, CA, USA) [25]. The volatile compounds were isolated using an HP-5MS quartz capillary column (0.25 µm, J&W Scientific Co., Ltd., Folsom, CA, USA) [26]. Helium was used as the column carrier gas at a flow rate of 1.0 mL/min. The initial temperature was controlled at 40 °C for 2 min, increased to 300 °C at a rate of 6 °C/min, and held at 300 °C for 5 min. Mass spectrometry parameters were set as follows: the electron ionization (EI) energy was 70 eV; the temperatures of the interface, quadrupole, and ion source were set at 280, 150, and 230 °C, respectively; and EI mass spectra ranged from 35 to 550 *m*/*z*. The qualitative and quantitative analysis of the sample was achieved by analyzing the mass charge ratio of the sample ions. The National Institute of Standards and Technology (NIST) Library database was used for spectrogram analysis [27]. The quantitative analysis of each volatile compound was calculated by the peak area normalization method [28].

### 2.6. Statistical Analysis

The design of RSM and optimization of the polynomials were performed using Design-Expert software, version 8.0. All significant differences between samples were evaluated using analysis of variance (ANOVA) using SPSS statistical 20.0 (SPSS Inc., Chicago, IL, USA), and the significance level was set to *p* < 0.05. All the treatments were carried out in triplicate, and results are expressed as the mean ± standard deviation (SD).

## 3. Results and Discussion

### 3.1. The Green Jujube Vinegar Process Optimization

#### 3.1.1. RSM Model for Total Acidity Content

The variance analysis of the regression RSM model and the coefficients in the equation are shown in Table 2. The model had a high F value (28.59) and a low *p*-value (*p* < 0.001), which indicated that the model was highly significant. The fitted quadratic polynomial regression model for the total acidity content (Equation (1)) was evaluated by the RSM and had a high R^2^ (0.9662). This indicates that the model has a better estimation accuracy for total acidity content in green jujube vinegar and can be used as the best estimation model between the factor and the response value [21].
Y (total acidity content (g/100 mL)) = 5.25 − 0.057 A − 0.053 B − 0.022 C − 0.15 D − 0.13 A B + 0.012 A C + 0.055 A D + 0.070 B C − 0.03 B D + 0.11 C D − 0.31 A^2^ − 0.31 B^2^ − 0.28 C^2^ − 0.28 D^2^(1)

#### 3.1.2. RSM Analysis and Verification of the Optimum Fermentation Parameters

The total acidity content has a direct impact on the quality of fruit vinegar and is an important indicator with which to verify the good character of fruit vinegar in China [29]. In Figure 1a–f, the three-dimensional response surface plots describe the interaction effect of the fermentation parameters. In response to the total acidity, the one-time item inoculation amount of acetobacter (*A*), fermentation temperature (*B*), and alcohol content (*D*), interaction terms *AB* and *CD*, and quadratic terms *A*^2^, *B*^2^, *C*^2^, and *D*^2^ were significant (*p* < 0.05). The other items had no significant differences (*p* > 0.05). 

Through the analysis of the main factor effect, the four factors selected for the experiment affected the total acidity content in the following order: alcohol content > inoculation amount of acetobacter > fermentation temperate > fermentation time. It was shown that fermentation is the result of the interaction of multiple factors, and the change trends between the four factors and the total acidity content (*Y*) were parabolic.

In general fruit vinegar practices, the maximization of the total acidity content is expected [29]. The optimum fermentation parameters for green jujube vinegar obtained from the model for initial alcohol content, acetic acid bacteria inoculation amount, fermentation temperature, and fermentation time were 6%, 8%, 32 °C, and 7 d, respectively. The content of residual sugars in the final products was 0.83 g/100 mL. Furthermore, the total acidity content of the green jujube vinegar was 5.38 g/100 mL under these conditions. To verify the accuracy of the test results, three parallel verification tests were carried out under the best process conditions. The total acidity content of the green jujube vinegar was 5.23 g/100 mL, which was slightly different from the predicted value. This suggested that the predicted values of this optimum condition were in good agreement with the actual measured values, which further demonstrated that the model can accurately predict the experimental results. It was concluded that the RSM is an effective method to optimize parameters such as initial alcohol content, acetic acid bacteria inoculation amount, fermentation temperature, and fermentation time. 

### 3.2. Analysis of Organic Acids during the Fermentation of Green Jujube Vinegar by HPLC

Organic acids are the components that most significantly influence the volatility and taste of fruit vinegar [29]. Therefore, the individual compounds of organic acids and their variation during the fermentation of green jujube vinegar were further analyzed in detail (Figure 2a–f). By HPLC, seven main organic acid compounds, including oxalic acid, tartaric acid, lactic acid, acetic acid, citric acid, malic acid, and succinic acid, were identified in fresh jujube juice (Figure 2a), alcohol fermented for 7 days (Figure 2b), acetic acid fermented for 7 days (Figure 2c), green jujube vinegar aged for 30 days (Figure 2d), and green jujube vinegar aged for 60 days (Figure 2e). The analysis of the total organic acid changes is summarized in Figure 2f.

The lactic acid and acetic acid concentrations increased significantly, from 0.077 g/100 mL and 0.096 g/100 mL in the fresh jujube juice to 1.267 g/100 mL (16.5-fold, *p* < 0.01) and 1.131 g/100 mL (11.8-fold, *p* < 0.01), respectively, in the green jujube vinegar aged for 30 days, and then decreased to 1.211 g/100 mL and 1.001 g/100 mL in the green jujube vinegar aged for 60 days. The succinic acid also significantly increased by 8.7-fold (from 0.055 g/100 mL to 0.478 g/100 mL, *p* < 0.01) during the fermentation process. By contrast, the tartaric acid, citric acid, and malic acid showed different variation trends during the fermentation process. The three acids dropped significantly (*p* < 0.05), decreasing from 0.070 g/100 mL, 0.212 g/100 mL, and 1.135 g/100 mL in the fresh jujube juice to 0.032 g/100 mL, 0.020 g/100 mL, and 0.042 g/100 mL, respectively, in the green jujube vinegar aged for 60 days. The malic acid content decreased the most significantly. The oxalic acid content increased during the alcohol fermentation (from 0.063 g/100 mL to 0.194 g/100 mL) and then decreased during the acetic acid fermentation and aging (from 0.194 g/100 mL to 0.050 g/100 mL) (Figure 2f). Overall, the green jujube vinegar (aged for 60 days) contained the following organic acids (in decreasing order of molar concentration): acetic acid > lactic acid > succinic acid > oxalic acid > malic acid > citric acid > tartaric acid.

Overall, the concentrations of the organic acids changed significantly during the alcohol fermentation, acetic acid fermentation, and aging. The changes were mainly attributed to microbial metabolism. The fermentation conditions might be the most important reason for the change in microorganisms’ metabolic activity. As the fermentation progresses, bioheat, alcohol, acidity, and enzymes have different effects on microorganisms at different stages of fermentation, resulting in various metabolic capacities in organic acids [30].

The concentrations of acetic acid and lactic acid increased significantly in the process of alcohol fermentation, acetic acid fermentation, and aging for one month. The synthesis of acetic acid and lactic acid involves many microorganisms, enzymes, and reactions. At the first stage of fermentation (alcohol fermentation), the increase in lactic acid content was mainly due to *Lactobacillus* metabolism and Saccharomycetes secondary metabolism. At the same time, the metabolism of *Lactobacillus* also produced a small amount of acetic acid. At the second stage of fermentation (acetic acid fermentation), the metabolism of *Lactobacillus* and Saccharomycetes was inhibited due to the environmental pressure of ethanol concentration, and *Acetobacter* played a leading role in the metabolism of the acetic acid, mainly through the pathways of acetyl-CoA, acetyl-adenylate, and acetaldehyde [31]. 

The content of succinic acid gradually increased during fermentation and aging, which was due to the transformation of the glutamic acid in the jujube, secondary metabolites of yeast, an intermediate product of glycometabolism, and a degradation product of protein. The concentrations of tartaric acid, citric acid, and malic acid constantly decreased during the fermentation and aging processes. The reduction in the tartaric acid concentration was due to the tartrate dehydratase of the lactic acid bacteria, which can convert tartaric acid into oxaloacetic acid and, next, into lactic acid, acetic acid, and CO_2_ [32]. The reduction in the citric acid concentration was a result of its decomposition into various products (such as diacetyl, lactic acid, and acetic acid) conducted by microbial strains [33]. The reduction in the malic acid concentration was the result of MLF (malic acid–lactic acid fermentation) and the esterification reaction of malic acid alcohols. In MLF, the malic acids in green jujube vinegar with lactic acid bacteria are converted into lactic acids and CO_2_, which is helpful for the improvement of green jujube vinegar quality [29]. The esterification reaction of malic acid alcohols endows vinegar with special aromatic components.

### 3.3. Identification and Analysis of Volatile Compounds by HS-SPME and GC-MS

The volatile compound was the direct influencing factor in the sensory evaluation of the fruit vinegar and was also an important index in the evaluation of the quality of the fruit vinegar. The changes in volatile components in fruit vinegar are usually related to a series of chemical reactions and microbial metabolism during fermentation and aging. In this study, the volatile substances in the fermentation and aging of green jujube vinegar were identified and analyzed by HS-SPME and GC-MS, and the total ion chromatograms are shown in Figure 3a–e. A total of 61 compounds were identified and quantified in the fermentation and aging processes, including 6 alcohols, 6 acids, 8 aldehydes, 27 esters, 6 hydrocarbons and derivatives, 3 ketones, and 6 other volatile compounds. 

In the fresh jujube juice (F0, 0 days), 30 volatile compounds were identified with a total concentration of 32.731 mg/L. In the alcohol fermentation (F1, 7 days) and acetic acid fermentation (F2, 14 days), the numbers of volatile components were 40 (total concentration 60.392 mg/L) and 41 (total concentration 50.413 mg/L), respectively. In the vinegar aged for 1 month (F3, 44 d) and 2 months (F4, 74 d), the numbers of volatile components were 39 (total concentration 62.902 mg/L) and 40 (total concentration 66.127 mg/L), respectively. The GC-MS analysis results of the total volatile compounds are shown in Table 3.

Alcohols. Six alcohols were detected as the major flavor compounds in the fermented green jujube vinegar: ethanol, isopentenyl, benzyl alcohol, 2-methyl-1-butanol, linalool, 4-terpineol, and dimethyl-silane-diol. The alcohols were mainly formed in the alcohol fermentation process; the degradation of amino acids, carbohydrates, and lipids probably imparts fusel, floral, and grass flavors [24,34,35]. The alcohols were the substrates of the acetic acid fermentation, and the remaining alcohols contributed desirable aromas to the jujube vinegar.

Acids. The volatile acids in the vinegar were the main flavor substances, and their unique taste and aroma, together with other flavor substances, constituted the characteristic flavor of the fruit vinegar [36]. Five volatile acid compounds, namely, isobutyric acid, acetic acid, hexanoic acid, 2-methylbutanoic acid, and isovaleric acid, were identified in this study. The volatile acids were an important index with which to evaluate the quality of the fruit vinegar, which was produced during fermentation and aging. Acetic acid was the characteristic flavor component of the jujube vinegar, which had a large change range of 0–26.165 mg/L. The content of the other acids in the jujube vinegar was relatively low. Appropriate concentrations of volatile acids can make fruit vinegar fresh, elegant, and pleasant, thus improving its quality. However, volatile acid contents in fruit vinegar that are excessively high are not conducive to fruit vinegar quality [36,37].

Aldehydes. Eight aldehydes were identified during the fermentation and aging of the green jujube vinegar: hexanal, (E)-2-hexenal, benzaldehyde, phenylethanal, trans-2-octenal, n-decanal, isovaleraldehyde, and 3-methyl-butanal. Six aldehyde compounds were detected in F0 with a total concentration of 3.996 mg/L, and five aldehyde compounds were detected in F1 with a total concentration of 6.773 mg/L, while three aldehyde compounds were detected in F4 with a total concentration of 0.736 mg/L. Aldehydes are known to be responsible for unpleasant smells. However, some aldehydes are beneficial to the aroma of vinegar, such as the green leaf aroma of trans-2-hexenal and the cherry aroma of benzaldehyde. In this study, the aldehydes of the green jujube vinegar decreased significantly during aging, from 6.773 mg/L to 0.736 mg/L. Most aldehydes are produced through lipid oxidation and the β-oxidation of free fatty acids by microbial fermentation and aldehydes [24]. They are unstable, which means they may be transformed into other volatile compounds. Acetaldehyde has been identified among the major aromatic compounds of many wines [38].

Esters. Esters are quantitatively the most abundant volatile compounds and are key aromatic compounds in many fruits and derivatives [39]. Esters are generally formed from the reaction between alcohol and acid compounds. In this study, twenty-seven esters were identified during the fermentation and aging of the green jujube vinegar (Table 3), making them the second most abundant volatile compounds in the contents. During the fermentation and aging, the esters of five samples (F0, F1, F2, F3, and F4) significantly differed in terms of their compositions and concentrations (*p* < 0.05). In the green jujube vinegar (F2, F3, and F4), isoamyl acetate was identified as the most dominant and highest in value (maximum value 16.422 mg/L), followed by ethyl acetate, methyl palmitate, methyl palmitoleate, 2-methyl butyl acetate, 2-methyl-1-butanol acetate, acetic acid-methyl ester, methyl elaidate, and isoamyl phenylacetate. Ubeda et al. [40] reported that isoamyl acetate (fruity and banana notes) was the most prominent ester in strawberry vinegar [35]. Ozen et al. and Ozdemir et al. reported isoamyl acetate, methyl palmitate, and methyl palmitoleate in sour cherry and hawthorn wines, respectively [38,41]. In addition to these, the esters detected in the present study were also found in sour cherry, hawthorn, grape, apple, pomegranate, and lemon vinegar. When compared to previous studies, the esters found in the F3 and F4 demonstrated higher concentrations, which showed a fruitier, more floral, and sweeter character.

In brief, green jujube vinegar includes a total of 61 volatile compounds that are associated with its aroma. The major volatile compounds include esters, acids, alcohols, siloxane, and aldehydes. The aromatic substances of the green jujube vinegar with the highest concentrations were acetic acid, isoamyl acetate, and ethyl acetate, three key compounds. Furthermore, 3-hydroxy-2-butanone, methyl palmitate, ethanol, 2-methyl-1-butanol, and methyl palmitate were important volatile aroma compounds. Acids and alcohols are not only primary metabolites but are also precursors of other aromatic compounds. Esters may undergo hydrolysis and other reactions to produce alcohol. Small amounts of volatile compounds are produced from other volatiles, which are secondary metabolites, such as ketones, aldehydes, or other substances. Although the concentrations of other volatiles were low, they played an important role in the green jujube vinegar aromas.

## 4. Conclusions

Organic acids are the key flavor compounds of green jujube vinegar, and their changes were mainly attributed to microbial metabolism. The inoculation amount and fermentation temperate of acetic acid bacteria are two key factors that result in different acid production. The fermentation parameters optimized by the RSM provide a feasible report for the production of green jujube vinegar. The organic acids of the optimized sample were evaluated by HPLC, and volatile substances were identified and analyzed by HS-SPME and GC-MS during the fermentation and aging of the green jujube vinegar. The statistical analysis showed that green jujube vinegar has rich organic acids and desirable volatile aromatic compounds, and that green jujube usage is ideal for vinegar production. Thus, green jujubes can be used as alternative products that can add value to the food sector and the world economy. Nevertheless, further studies should be performed to explore the relationship between the core microbes and the flavor compounds, and the specific changes of the polyphenols and flavonoids during the fermentation and aging of green jujube vinegar need to be further investigated.

## Figures and Tables

**Figure 1 foods-12-03168-f001:**
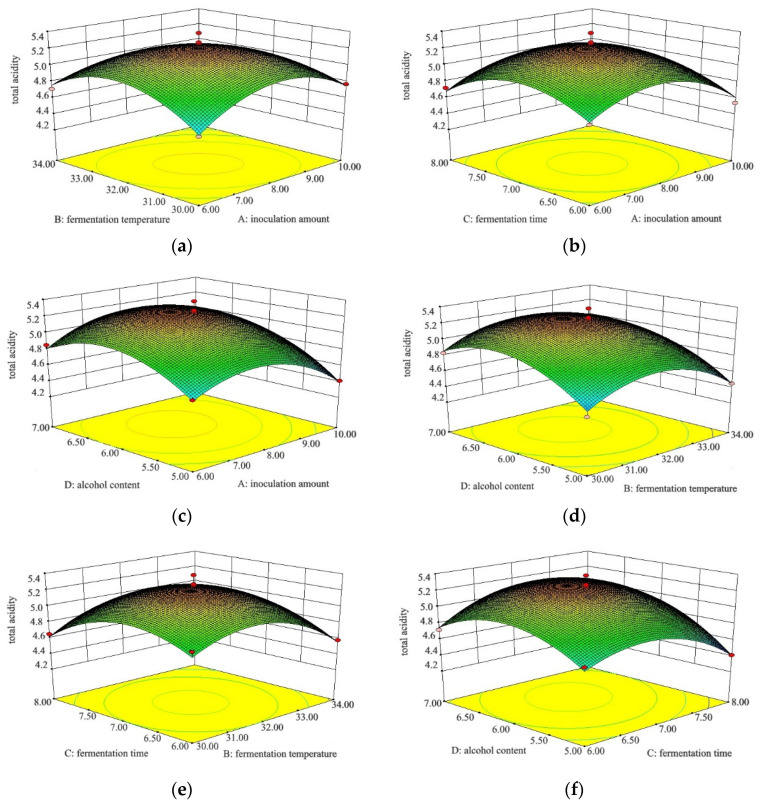
Response surface 3D diagram of the effects of four independent variables on total acidity of green jujube vinegar. A, B, C, and D represent inoculation amount of acetobacter (%, *v*/*v*), fermentation temperature (°C), fermentation time (d), and alcohol content (%, *v*/*v*), respectively. (**a**) A and B are two variables, C = 7.00 h, and D = 6.00%; (**b**) A and C are two variables, B = 32.00 °C, and D = 6.00%; (**c**) A and D are two variables, B = 32.00 °C, and C = 7.00 h; (**d**) B and D are two variables, A = 8.00%, and C = 7.00 h; (**e**) B and C are two variables, A = 8.00%, and D = 6.00%; and (**f**) C and D are two variables, A = 8.00%, and B = 32.00 °C.

**Figure 2 foods-12-03168-f002:**
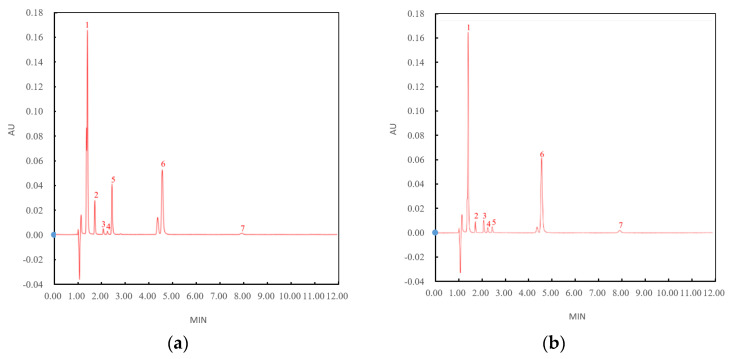
Analysis of organic acids during the fermentation of green jujube vinegar by HPLC, (1) oxalic acid, (2) tartaric acid, (3) lactic acid, (4) acetic acid, (5) citric acid, (6) malic acid, and (7) succinic acid; (**a**) fresh jujube juice, (**b**) alcohol fermented for 7 days, (**c**) acetic acid fermented for 7 days, (**d**) green jujube vinegar aged for 1 month, (**e**) green jujube vinegar aged for 2 months, and (**f**) analysis of total organic acids in the fermentation of green jujube vinegar.

**Figure 3 foods-12-03168-f003:**
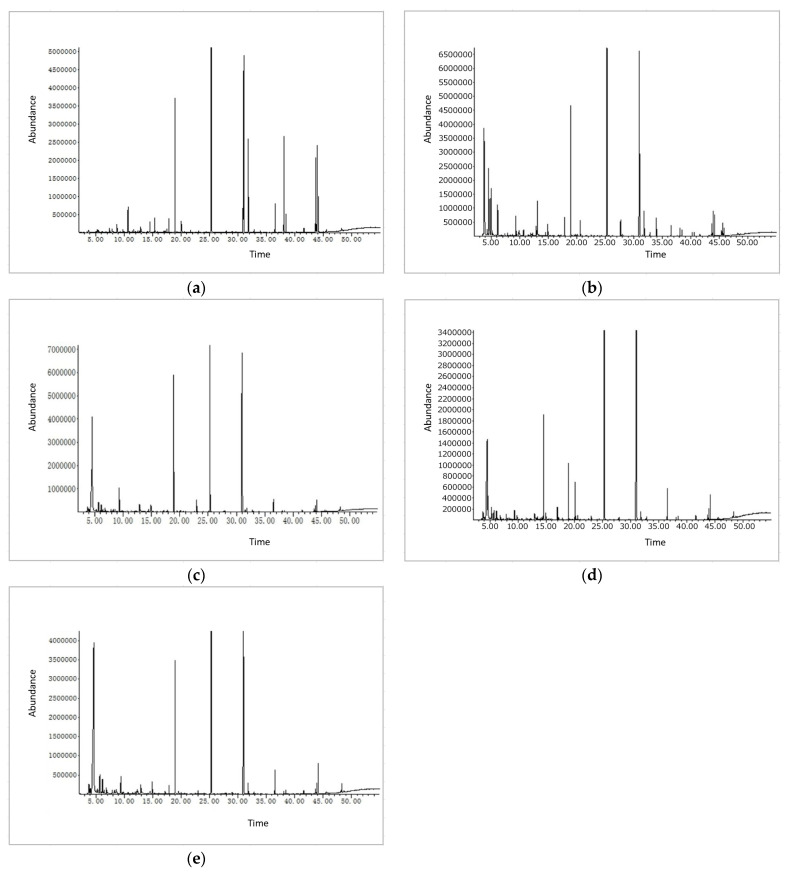
Analysis of flavor compounds during fermentation of green jujube vinegar by HS-SPME-GC-MS, (**a**) fresh jujube juice, (**b**) alcohol fermentation for 7 days, (**c**) acetic acid fermentation for 7 days, (**d**) green jujube vinegar aged for 1 month, (**e**) green jujube vinegar aged for 2 months.

**Table 1 foods-12-03168-t001:** Design of the response surface methodology with independent variables and modeled data responses.

Run	A	B	C	D	Y
1	−1	0	0	−1	4.63 ± 0.16
2	+1	−1	0	0	4.77 ± 0.53
3	0	0	+1	+1	4.85 ± 0.19
4	0	0	0	0	5.25 ± 0.09
5	0	+1	0	+1	4.79 ± 0.03
6	+1	+1	0	−1	4.38 ± 0.37
7	0	0	−1	−1	4.71 ± 0.11
8	0	−1	+1	−1	4.65 ± 0.16
9	0	0	0	0	5.38 ± 0.05
10	−1	0	0	0	4.59 ± 0.53
11	+1	−1	0	−1	4.50 ± 0.23
12	+1	0	−1	0	4.54 ± 0.38
13	+1	0	0	+1	4.85 ± 0.73
14	0	0	0	0	5.20 ± 0.09
15	+1	0	0	−1	4.41 ± 0.03
16	−1	0	0	+1	4.85 ± 0.17
17	−1	0	0	0	4.72 ± 0.62
18	0	+1	0	−1	4.45 ± 0.08
19	0	0	0	0	5.13 ± 0.05
20	0	0	−1	+1	4.72 ± 0.70
21	0	0	+1	−1	4.41 ± 0.63
22	0	0	0	0	5.27 ± 0.05
23	0	+1	−1	0	4.58 ± 0.16
24	−1	+1	0	0	4.71 ± 0.41
25	0	−1	0	+1	4.83 ± 0.32
26	−1	0	+1	0	4.72 ± 0.27
27	1	0	+1	0	4.59 ± 0.11
28	0	1	+1	0	4.65 ± 0.05
29	0	−1	−1	0	4.86 ± 0.21

Note: A, inoculation amount of acetobacter (%); B, fermentation temperate (°C); C, fermentation time (d); D, initial alcohol content (%, *v*/*v*); and Y, total acidity content (g/100 mL).

**Table 2 foods-12-03168-t002:** Analysis of variance of the regression RSM model.

Source	SS	DF	MS	F-Value	*p*-Value	Significance	R^2^	R^2^ Adj
Model	1.93	14	0.14	28.59	<0.0001	**	0.9662	0.9324
A	0.039	1	0.039	7.99	0.0135	*		
B	0.034	1	0.034	7.08	0.0186	*		
C	5.633 × 10^−3^	1	5.633 × 10^−3^	1.17	0.2981			
D	0.26	1	0.26	54.75	<0.0001	**		
A B	0.065	1	0.065	13.48	0.0025	*		
A C	6.250 × 10^−4^	1	6.250 × 10^−4^	0.13	0.7242			
A D	0.012	1	0.012	2.51	0.1355			
B D	2.500 × 10^−5^	1	2.500 × 10^−5^	5.184 × 10^−3^	0.9436			
C D	0.046	1	0.046	9.58	0.0079	**		
A^2^	0.62	1	0.62	128.97	<0.0001	**		
B^2^	0.62	1	0.62	128.97	<0.0001	**		
C^2^	0.51	1	0.51	105.19	<0.0001	**		
D^2^	0.51	1	0.51	105.19	<0.0001	**		
Residual	0.068	14	4.823 × 10^−3^					
Lack of Fit	0.033	10	3.340 × 10^−3^	0.39	0.8955			
Pure Error	0.034	4	8.530 × 10^−3^					
Synthesis	2.00	28						

A, B, C, and D represent inoculation amount of acetobacter (%, *v*/*v*), fermentation temperature (°C), fermentation time (d), and alcohol content (%, *v*/*v*), respectively; SS, sum of squares; DF, degree of freedom; MS, mean square; and “*” represents significant difference (*p* < 0.05). “**” represents highly significant difference (*p* < 0.01).

**Table 3 foods-12-03168-t003:** Analysis of volatile compounds in the fermentation of green jujube vinegar.

No.	Volatile Compounds	CAS	Formula	MW	Concentration (mg/L)				
					F0	F1	F2	F3	F4
	**6 alcohols**								
C1	Ethanol	64-17-5	C_2_H_6_O	46.1	0.273 ± 0.031 ^c^	16.396 ± 1.235 ^a^	4.949 ± 0.236 ^b^	2.113 ± 0.167 ^b^	1.313 ± 0.124 ^bc^
C2	Isopentanol	123-51-3	C_5_H_12_O	88.2	0.112 ± 0.014 ^c^	2.153 ± 0.117 ^b^		0.171 ± 0.051 ^c^	
C3	Benzyl alcohol	100-51-6	C_7_H_8_O	108.1					0.196 ± 0.061 ^c^
C4	2-Methyl-1-butanol	137-32-6	C_5_H_12_O	88.2		2.330 ± 0.207 ^b^	0.785 ± 0.036 ^c^	0.655 ± 0.034 ^c^	1.022 ± 0.087 ^c^
C5	Linalool	78-70-6	C_10_H_18_O	154.3				0.647 ± 0.031 ^c^	
C6	4-Terpineol	562-74-3	C_10_H_18_O	154.3	0.555 ± 0.034 ^c^	0.120 ± 0.025 ^c^	0.125 ± 0.064 ^c^	1.920 ± 0.238 ^bc^	
	**5 acids**								
C7	Isobutyric acid	79-31-2	C_4_H_8_O_2_	88.2			0.271 ± 0.013 ^c^	0.156 ± 0.055 ^c^	0.299 ± 0.041 ^c^
C8	Acetic acid	64-19-7	C_2_H_4_O_2_	60.1		0.905 ± 0.015 ^c^	11.450 ± 0.465 ^b^	22.457 ± 2.241 ^a^	26.165 ± 2.005 ^a^
C9	Hexanoic acid	142-62-1	C_6_H_12_O_2_	116.2			0.213 ± 0.028 ^c^		
C10	2-Methylbutanoic acid	116-53-0	C_5_H_10_O_2_	102.1			0.304 ± 0.025 ^c^	0.122 ± 0.024 ^c^	0.434 ± 0.123 ^c^
C12	Isovaleric acid	503-74-2	C_5_H_10_O_2_	102.1			0.419 ± 0.033 ^c^	0.163 ± 0.015 ^c^	0.339 ± 0.021 ^c^
	**8 aldehydes**								
C13	Hexanal	66-25-1	C_6_H_12_O	100.2	0.391 ± 0.052 ^b^	0.223 ± 0.105 ^b^			
C14	(E)-2-Hexenal	6728-26-3	C_6_H_10_O	98.1	0.767 ± 0.036 ^b^				
C15	Benzaldehyde	100-52-7	C_7_H_6_O	106.1	1.714 ± 0.122 ^b^	0.384 ± 0.062 ^b^			0.239 ± 0.231 ^b^
C16	Phenylethanal	122-78-1	C_8_H_8_O	120.2		0.997 ± 0.065 ^b^	0.831 ± 0.153 ^b^	0.578 ± 0.063 ^b^	0.054 ± 0.002 ^c^
C17	trans-2-Octenal	2548-87-0	C_8_H_14_O	126.2	1.073 ± 0.232 ^b^				
C18	n-Decanal	112-31-2	C_10_H_20_O	156.3	0.051 ± 0.009 ^c^	0.048 ± 0.019 ^c^	0.045 ± 0.006 ^c^		
C19	Isovaleraldehyde	590-86-3	C_5_H_10_O	86.1		5.121 ± 0.931 ^a^			
C20	3-Methyl-Butanal	590-86-3	C_5_H_10_O	86.1				0.251 ± 0.013 ^c^	0.443 ± 0.034 ^c^
	**27 esters**								
C21	Isoamyl formate	110-45-2	C_6_H_12_O_2_	116.2			0.183 ± 0.012 ^c^		0.171 ± 0.013 ^c^
C22	Isoamyl acetate	102-19-2	C_7_H_14_O_2_	130.2	8.511 ± 0.335 ^ab^	11.672 ± 0.765 ^a^	13.874 ± 1.521 ^a^	15.707 ± 1.876 ^a^	16.422 ± 3.532 ^a^
C23	Methyl dodecanoic	111-82-0	C_14_H_28_O_3_	244.4	0.583 ± 0.053 ^c^	1.470 ± 0.065 ^b^	0.317 ± 0.042 ^c^	0.469 ± 0.003 ^c^	0.584 ± 0.013 ^c^
C24	Ethyl laurate	106-33-2	C_14_H_28_O_2_	228.4	0.129 ± 0.087 ^c^	1.056 ± 0.023 ^b^	0.046 ± 0.009 ^c^		
C25	Acetic acid-methyl ester	79-20-9	C_3_H_6_O_2_	74.1			0.382 ± 0.05 ^c^	0.392 ± 0.003 ^c^	0.704 ± 0.234 ^bc^
C26	Ethyl acetate	141-78-6	C_4_H_8_O_2_	88.1		0.857 ± 0.098 ^bc^	3.487 ± 0.243 ^b^	4.896 ± 0.125 ^b^	6.123 ± 0.053 ^ab^
C27	Phenethyl acetate	103-45-7	C_10_H_12_O_2_	164.2			1.211 ± 0.034 ^b^		0.181 ± 0.005 ^c^
C28	Isobutyl acetate	110-19-0	C_6_H_12_O_2_	116.2		0.121 ± 0.032 ^c^	0.369 ± 0.030 ^c^	0.269 ± 0.006 ^c^	0.382 ± 0.004 ^c^
C29	2-Methylbutyl acetate	624-41-9	C_7_H_14_O_2_	130.2		0.273 ± 0.062 ^c^	0.904 ± 0.018 ^bc^	0.457 ± 0.022 ^c^	0.831 ± 0.032 ^bc^
C30	Methyl octanoate	111-11-5	C_9_H_18_O_2_	158.2	0.901 ± 0.112 ^bc^	1.064 ± 0.043 ^b^		0.121 ± 0.006 ^c^	0.468 ± 0.043 ^c^
C31	Ethyl octanoate	106-32-1	C_10_H_20_O_2_	172.3		0.871 ± 0.085 ^bc^			0.066 ± 0.037 ^c^
C32	Methyl oleate	112-62-9	C_19_H_36_O_2_	296.5				0.129 ± 0.002 ^c^	
C33	Ethyl decanoate	110-38-3	C_12_H_24_O_2_	200.3		0.863 ± 0.064 ^bc^			0.058 ± 0.033 ^c^
C34	Methyl myristoleate	56219-06-8	C_15_H_28_O_2_	240.4	0.692 ± 0.098 ^bc^	0.620 ± 0.007 ^c^	0.160 ± 0.009 ^c^		0.169 ± 0.023 ^c^
C35	Methyl myristate	124-10-7	C_15_H_30_O_2_	242.4	1.248 ± 0.085 ^b^	0.408 ± 0.001 ^c^	0.152 ± 0.000 ^c^	0.204 ± 0.025 ^c^	0.232 ± 0.065 ^c^
C36	Ethyl myristate	124-06-1	C_16_H_32_O_2_	256.4		0.190 ± 0.000 ^c^			
C37	Methyl palmitoleate	1120-25-8	C_17_H_32_O_2_	268.4	3.647 ± 0.014 ^b^	2.165 ± 0.034 ^b^	0.781 ± 0.005 ^c^	0.840 ± 0.053 ^c^	0.811 ± 0.018 ^bc^
C38	Methyl palmitate	112-39-0	C_17_H_34_O_2_	270.5	2.210 ± 0.059 ^b^	1.223 ± 0.063 ^b^	0.925 ± 0.013 ^c^	1.209 ± 0.024 ^b^	1.480 ± 0.085 ^b^
C39	Ethyl palmitate	628-97-7	C_18_H_36_O_2_	284.5		0.445 ± 0.009 ^c^	0.067 ± 0.003 ^c^	0.054 ± 0.006 ^c^	
C40	Ethyl palmitoleate	56219-10-4	C_18_H_34_O_2_	282.5		1.101 ± 0.063 ^b^			
C41	Methyl caproate	106-70-7	C_7_H_14_O_2_	130.2	1.515 ± 0.042 ^b^	0.358 ± 0.005 ^c^			0.071 ± 0.006 ^c^
C42	Ethyl caproate	123-66-0	C_8_H_16_O_2_	144.2		1.941 ± 0.086 ^b^	0.181 ± 0.013 ^c^	0.164 ± 0.033 ^c^	0.162 ± 0.042 ^c^
C43	Methyl phenylacetate	101-41-7	C_9_H_10_O_2_	150.2	0.826 ± 0.098 ^bc^	0.107 ± 0.005 ^c^	0.095 ± 0.000 ^c^	0.151 ± 0.033 ^c^	0.086 ± 0.036 ^c^
C44	Ethyl benzoate	93-89-0	C_9_H_10_O_2_	150.2		0.139 ± 0.004 ^c^			
C45	Methyl linoleate	112-63-0	C_19_H_34_O_2_	294.5			0.070 ± 0.013 ^c^		0.070 ± 0.007 ^c^
C46	Methyl elaidate	1937-62-8	C_19_H_36_O_2_	296.5	0.228 ± 0.063 ^c^	0.153 ± 0.032 ^c^	0.501 ± 0.002 ^c^	0.364 ± 0.009 ^c^	0.485 ± 0.074 ^c^
C47	2-Methyl-1-butanol acetate	624-41-9	C_7_H_14_O_2_	130.2			0.304 ± 0.013 ^c^	0.852 ± 0.63 ^bc^	0.831 ± 0.054 ^bc^
	**6 hydrocarbons and derivatives**								
C48	Hexamethyl- cyclotrisiloxane	541-05-9	C_6_H_18_O_3_Si_3_	222.5	0.323 ± 0.006 ^ab^	0.275 ± 0.003 ^ab^	0.311 ± 0.005 ^ab^	0.478 ± 0.007 ^ab^	
C49	Octamethyl-cyclotetrasiloxane	556-67-2	C_8_H_24_O_4_Si_4_	296.6	0.502 ± 0.009 ^ab^	0.640 ± 0.005 ^b^	0.731 ± 0.004 ^b^		0.641 ± 0.086 ^b^
C50	Tetradecamethyl-cycloheptasiloxane	107-50-6	C_14_H_42_O_7_Si_7_	519.1	1.304 ± 0.132 ^a^	0.491 ± 0.052 ^ab^	0.515 ± 0.021 ^ab^	0.647 ± 0.016 ^b^	0.748 ± 0.009 ^b^
C51	Octadecamethyl-cyclopentasiloxane	556-71-8	C_18_H_54_O_9_Si_9_	667.4	0.300 ± 0.015 ^c^	0.113 ± 0.009 ^c^	0.173 ± 0.002 ^c^	0.241 ± 0.002 ^c^	0.208 ± 0.015 ^c^
C52	Eicosamethyl-cyclopentasiloxane	18772-36-6	C_20_H_60_O_10_Si_10_	741.5				0.147 ± 0.022 ^c^	0.106 ± 0.003 ^c^
C53	3, 4-Dihydroxyphenylglycol-4TMS derivative	56114-62-6	C_20_H_42_O_4_Si_4_	458.9	2.182 ± 0.132 ^a^	0.755 ± 0.086 ^b^	1.244 ± 0.160 ^a^	1.831 ± 0.87 ^a^	
	**2 ketones**								
C54	3-Hydroxy-2-butanone	513-86-0	C_4_H_8_O_2_	88.1			0.181 ± 0.012 ^c^	0.849 ± 0.014 ^b^	1.589 ± 0.075 ^a^
C55	1-Octen-3-one	4312-99-6	C_8_H_14_O	126.2	0.118 ± 0.004 ^c^				
	**6 others**								
C56	Eucalyptol	470-82-6	C_10_H_18_O	154.3	0.680 ± 0.036 ^b^		0.251 ± 0.007 ^c^	0.498 ± 0.012 ^bc^	
C57	7-Hydroxytotarol, di(trimethylsilyl) ether	1000386-45-3	C_18_H_38_	254.5			0.212 ± 0.006 ^c^		0.157 ± 0.025 ^c^
C58	2-Phenylcyclopropionamid-e, N-(4phenylazo)phenyl-	303097-62-3	C_22_H_19_N_3_O	341.4		0.159 ± 0.002 ^c^			
C59	Ammonium acetate	631-61-8	C_2_H_7_NO_2_	77.1			0.205 ± 0.006 ^c^		
C60	Silane, methyl vinyl(2-methylpent-3-yloxy)(methylvinyldodec-yloxysilyloxy)-	1000421-54-9	C_11_H_24_O_2_Si	216.4	0.357 ± 0.009 ^bc^			0.192 ± 0.003 ^c^	
C61	Oxime-, methoxy-phenyl-	1000222-86-6	C_7_H_6_ClNO_2_	171.6	0.270 ± 0.007 ^c^				

Notes: F0 represents fresh jujube juice (0 d), F1 represents alcohol fermented for 7 days (7 d), F2 represents acetic acid fermented for 7 days (14 d), F3 represents green jujube vinegar aged for 1 month (44 d), and F4 represents green jujube vinegar aged for 2 months (74 d). ^abc^: Different letters indicate significant differences between fermentation days of the same sample (*p* < 0.05).

## Data Availability

Data are contained within the article.

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
