# Peer review of "Optimization of the Process for Green Jujube Vinegar and Organic Acid and Volatile Compound Analysis during Brewing"

_foods, 2023, doi:10.3390/foods12173168_

Round 1
Reviewer 1 Report (Previous Reviewer 1)
The method for the determination of organic acids states that 20 mmol/L potassium dihydrogen phosphate buffer is used as mobile phase A but does not specify the pH of the buffer, it is an important value for the method used.
In the model for estimating the acidity content, variables that are not significant have been used; they should be eliminated from the proposed model and only significant variables should be used. To check whether or not it influences the best estimation of the proposed model.
Figure 1 represents the surface curve when varying the value of two variables, but the value of the rest of the variables used to obtain this curve must be indicated. The nomenclature of the variables used in the previous table and model must be maintained.
For the quantified values of volatile compounds, the significant figures of the different values should be taken into account according to the error of the values.
In table 3, superscripts (a, b, c) are used for reference which are not detailed in the table footnote.
The text of the supplementary material is not in English.
Author Response
Please see the attachment.

Reviewer 2 Report (New Reviewer)
Line 79: This sentence needs to be further clarified by what you mean in regards to sugar water during 35-38C.
Line 118: Should this be during brewing and not in the brewing?
Line 147: I would put your GC column with the GC-MS section. It would make more sense since you are discuss the GC-MS.
Figure 2: The HPLC chromatographs are difficult to read the axis. The graph you developed honestly would be a better figure just on its own.
Line 311: 3-methylbutanoic acid and isovaleric acid are the same compound. This needs to be corrected in the manuscript and in the table.
Line 321: Double check your spelling for pheneylethanol. Was it supposed to be an "al" instead of "ol"?
In lieu of CAS numbers I would recommend adding the VOC data instead.
Do you feel that the siloxane derivatives that you found in your samples are a byproduct of the fermentation or associated with breakdown from your SPME fiber and column.
There are a few sentences that need to be reworked so that they can be made clearer for the reader.
Author Response
Please see the attachment.

Reviewer 3 Report (New Reviewer)
Dear Authors,
In general, the manuscript is good to read, the structure of the work is clear. After reading the paper, it is clear that the authors performed a lot of experiments and analyzes in order to obtain detailed research results. I present my comments below:
1. Supplementary materials are mostly described in Chinese. Please correct.
2. Line 70. Name the jujube variety.
3. Lines 171 - 175. These sentences are more suitable for describing the methodology.
4. Lines 191 - 192. And this statement fits more with Introduction.
5. "In general fruit vinegar practices, the maximization of the total acidity content is expected [29]". How does this statement relate to the manuscript about fish? The authors quote here the paper: "Identification of changes in volatile compounds in dry-cured fish during storage using HS-GC-IMS".
6. Figures 2 from (a) to (e) the axis description font is too small.
7. Lines 285 - 288. These two sentences are more suitable for Introduction.
8. Lines 368 - 373. These are not final conclusions.
In general, the work contains a lot of interesting experimental data. In my opinion, the Principal Components Analysis fits perfectly to the interpretation of the results. E.g. data from Table 3.
Author Response
Please see the attachment.

This manuscript is a resubmission of an earlier submission. The following is a list of the peer review reports and author responses from that submission.
Round 1
Reviewer 1 Report
All references used are from authors of Chinese nationality, sources of other nationalities should also be consulted.
The initial sugar content to alcoholic fermentation has been adjusting to 24%, the average sugar concentration present in green jujube juice should also be indicated.
The method for the determination of organic acids states that 20 mmol/L potassium dihydrogen phosphate buffer is used as mobile phase A but does not specify the pH of the buffer.
In the model for estimating the acidity content, variables that are not significant have been used; they should be eliminated from the proposed model and only significant variables should be used. To check whether or not it influences the best estimation of the proposed model.
Table 1 shows the design of the response surface methodology for run 9 with the values 0000 (8% inoculation amount, 7d fermentation time, 32ºC fermentation temperature and 6% alcohol content and obtains a value of total acidity content of 5.38 g/100mL and corresponds to the intermediate values of all variables, in line 104 the coded levels of one of the variables are missing. The error of the three parallel verificacion tests must be indicated in line 202, as well as, in table 1 the error in the determination of the total acidity content for each of the working conditions.
For all quantified values for organic acids and volatile compounds, the error of quantification must be indicated in order to determine whether the differences found between the different points F0, F1, F2, F3 and F4 for each of the compounds are significant or not.
Author Response
Dear Peer Reviewers and Editor,
How are you?
Thank you for your careful review of this article and valuable comments. We have made serious modifications and supplements according to your opinions. All the modifications have been marked in yellow or red in the revised draft.
For the attached amendments, we have revised and supplemented them item by item as follows:
English language and style
This article has been edited and polished by MDPI English Editing regarding the English language problem.
Comments and suggestions for authors
- All references used are from authors of Chinese nationality, sources of other nationalities should also be consulted.
Response 1: According to this comment, the references have been revised carefully. Twelve references (4, 5, 7, 8, 16, 17, 30, 32, 39, 40, 41, 42) are from authors of other nationalities.
- The initial sugar content to alcoholic fermentation has been adjusting to 24%, the average sugar concentration present in green jujube juice should also be indicated.
Response 2: The sugar concentration of green jujube juice is about 11%, the initial sugar content to alcoholic fermentation was adjusted to 24%. (Line 79)
- The method for the determination of organic acids states that 20 mmol/L potassium dihydrogen phosphate buffer is used as mobile phase A but does not specify the pH of the buffer.
Response 3: In this study, the pH of the potassium dihydrogen phosphate buffer is 4-5. (Line 128)
- In the model for estimating the acidity content, variables that are not significant have been used; they should be eliminated from the proposed model and only significant variables should be used. To check whether or not it influences the best estimation of the proposed model.
Response 4: In the model for estimating the acidity content, variables that not significant variables have been deleted (X3, X1 X3, and X2 X4). (Line 175-176)
- The error of the three parallel verification tests must be indicated in line 202, as well as, in table 1 the error in the determination of the total acidity content for each of the working conditions.
Response 5: Thanks for your valuable suggestion. I apologize for my negligence. The error of the three parallel verification tests has been supplemented in table 1 and line 202.
- For all quantified values for organic acids and volatile compounds, the error of quantification must be indicated in order to determine whether the differences found between the different points F0, F1, F2, F3 and F4 for each of the compounds are significant or not.
Response 6: Thanks for your valuable suggestion. I apologize for my negligence.
Now the error of quantification for organic acids and volatile compounds has been added (line 220-234 and table 3).

Reviewer 2 Report
Dear Authors
I have read and revised the manuscript titled "Optimization of the Process for Green Jujube Vinegar and Or-2 ganic Acids and Volatile Compounds Analysis during Winemaking". Although the manuscript surely has merits, there are too many flaws that require attention and have to be amended. As a general comment, the methology employed has merit, as the experimental design was well-conceived, however several flaws can be spotted in the methodological and analytical parts, both in the detailing various aspects of the activites don (e.g. inocula, types and strains of the yeast and bacteria used), both in the interpretation of the results and their presentation. Finally, although worth of attention, the lack of sensory characterization of the produced vinegars is a limit to the scope laid out in the introduction, so related changes to key parts are in order. Puntual comments are reported here below.
Kind Regards
Comments:
1) Title: please, check the English ("... and... and..." does not sound correct)
2) Abstract: The last sentence is indeed a bit too much. The lack of sensory data or a consumer survey limits the scope of the work, as it does not allow to asses the quality of the products produced, from a consumer perspective.
3) Introduction - Line 40: please remove the expression "and so on".
Material and Methods
4) Line 74: You should report here (even from literature data) the average content of pectin in the raw material (see following comment).
5) Line 74-76: Have you monitored the production of methanol in the final products? It is relevant in relation to pectinase activity.
6) Paragraph 2.2: This paragraph is really insufficient. More details are required on the applied yeasts and bacteria (strains, commercial names if any, preparation before inoculum...). Please, see also further notes.
7) Paragraph 2.2 - Line 78-82: Have you monitored the final amount of residual fermentabel sugars? How have you monitored the final amount ethanol produced? Refractometry cannot give an accurate estimation in presence of both.
8) Paragraph 2.2 - From Line 83: Please, add more details here on the conditions used for the inoculum both of alcoholic and acetic fermentations? What yeast did you use? What about the acetic bacteria? From the Results, it seems you also used Lactic bacteria... whichi ones, when? Please, add details.
9) Paragraph 2.2 - How did you measure the produced alcohol? As mentioned, refractometry does not work well to monitor fermentations when both sugars and alcohol are present.
10) Paragraph 2.2 - Line 88-89: Can you briefly detail here the method used for total acidity determination, especially for non-speakers of Chinese?
11) Table 1: I think the responses should be given space in the Results section. Maybe, you could just show here the experimental matrix. Besides, is this the total acidity after the ageing time?
12) Paragraph 2.4.2 - Line 123-124: Please, be careful how you write chemical names in lists. 1-oxalic acid, 2-tartaric acid etc. is a really unacceptable way to list chemicals. Please, change them to 1) oxalic acid, 2) tartaric acid...
13) Paragraph 2.5.2 - Line 149-150: "The qualification was achieved by com-149 parison of the mass spectra and retention times with standards"... did you calculate the retention indexes? Against which standards? Linear alkanes?
Results and Discussion
14) Paragraph 3.1.1: The authors should state much more clearly and probably also in te introduction beside also here the optimum they sought for total acidity (the lowest possible, the highest possible), otherwise it is really unclear why they applied this RSM method for this specific parameter. In relation to this, in the Introduciton you wrote "This study aimed to produce vinegar with strong functional properties from green jujubes and to convert this fruit into a high-value-added product." Well, was only acetic acid and lactic acid at the base for these "functional properties"? If so, you should consider explaining why, or change that part in the Introduction.
15) Paragraph 3.1.1: Exactly what do the authors mean by "optimizing the fermentation conditions?" Only increasing the total acidity? Increasing total acidity? Increasing specific organic acids production? The authors should state this more clearly.
16) Table 2: The RSM model is very nice indeed, however the authors scould clearly state the purpose of optimizing for the total acidity.
17) Paragraph 3.1.2: I think it is not ok separating the discusison from the Table presenting the model parameters, e.g. by creating a new paragraph.
18) Paragraph 3.1.2 - Line 183-184: This sentence is really insufficient. Please, provide any reference to regulatory sources, where the maximal/minimal values for these quality parameters in this vinegar are laid out.
19) Line 187: "temperate". Please, amend.
20) Line 187: please, uniform the listing using only commas or semicolons.
21) Line 195: Please, include a reference. Not all readers are expert on vinegar production.
22) Line 198: "Further" should be "Furthermore". Please, check.
23) Line 200: What do the authors mean with "authenticity"? Did they really mean the "accuracy" for model in prediction?
24) Figure 1: Either improve the quality of these Figures, of move to the Supporting Information.
25) Paragraph 3.2 - Line 212: What is "volatility"? Pehraps, did the authors mean "olfactory aroma profile" or "volatile aroma profile"? That would really mean (considering just the analyzed compounds in this paragraph) the only organic acid directly contributing to the volatile acidity... i.e. mostly acetic acid and to a lesser extent lactic acid.
26) Paragraph 3.2 - Line 235: Such comparison can be done, but only after conversion of these acids concentration to molarities. Please, check that the order is respected even considering molar concentrations.
27) Paragraph 3.2 - Line 237: A plot would be nice, highlighting the evolution of the organic acids at least for the optimal fermentation process, from the beginning (raw juice) to the end (aged vinegar).
28) Paragraph 3.2 - Can the Authors report the content of residual sugars in the final products?
29) Paragraph 3.2 - Line 249: The authors did not mention before an inoculum with Lactobacillus... they must include this in the Material and Methods. Were they added together with Acetobacter? All details abouth the inocula should be included there.
30) Figure 2: Please, improve graphic quality.
31) Figure 2 - caption: Please, see previous comment on listing chemical compounds.
32) Paragraph 3.3: In the entire section and in Table 3, please check the number of significant digits associated to concentration values. Are they all significant?
33) Line 282: "kinds of volatile compounds"... please, rephrase. Those were indeed "individually identified compounds".
34) Line 331: Why choosing only 5 samples? Besides, please see comment to Table 3.
34) Table 3: Were the concentrations (mg/L) obtained by semi-quantitation over the internal standard? If so, please specify it in the caption. Besides... the internal standard chioice is greatly flawed... see comment below. Besides, please indicate what F1... F5 are in the caption.
35) Table 3: You should really report the error estimations for the presented concentrations. Are "biological" replicates present?
36) Table 3: Please, add one column with the found base mass (m/z) and another with the measured retention index.
37) Table 3: are concentration values averages? Have technical replicated being
38) Table 3 - compound C7: Silanes and siloxanes are released from the chromatographic apparatus. Please, remove them from the Table.
39) Table 3 - compound C30: Isoamyl phenylacetate was used as the internal standard... So, its changing amount in the different samples indicate that it was not the correct choice.
40) Table 3 - C50:C56: All compounds released from the apparatus (column, SPME fiber)... these are all intereferents.
Round 2
Reviewer 2 Report
Dear Authors,
I have now read the revised version of the manuscript. Each point raised was addressed. However, two points would require additional care. The first one, concerns the content of methanol due to induced pectin degradation by pectinases. The GC-MS settings show the lowest mass detected at m/z 35 (please, see the reported reference EI MS spectrum of methanol in the NIST database). Clearly, these conditions are not suitable for detecting methanol. I suggest therefore at least to add a comment in the Results, indicating that careful monitoring of methanol production should be ensured in the final product and valuable countermeasures should be considered.
The second point regards the internal standard in the GC-MS analysis. Unfortunately it miight not be enough that you have removed it from the Table. It cannot be used in any way as an internal standard. Its variation across the samples indicates that it might well have been produced over the fermentation, from esterification of phenylacetic acid with isoamyl alcohol (both present in fermented beverages). I must therefore suggest to remove entirely the indication of the use of the internal standard from the Material and Methods, paragraph "2.5. analysis of volatile compounds". This has a further consequence, as the use of the internal standard for calculating concentrations (in Table 3) is no more possible in these circumstances. It is therefore necessary that in Table 3 relative abundances (peak areas) for each chemical compound are compared between the samples, instead of absolute concentrations (which are probably wrong, considering the flaw in IS choice). There would be also another possible option. Instead of the areas, the compounds' internal percentages (i.e. for one sample, it is the %-ratio of the area of each compound inside a chromatogram over the sum of areas of all peaks inside the same chromatogram, then reported in percentage) could be compared between samples, which might be a valuable alternative available that might work.
I think that these major modifications are unfortunately also necessary, in view of the impossibility of the use of isoamyl phenylacetate as IS. Otherwise, the value reported in Table 3 would clearly contain a source of systematic error, which is to be avoided.
